# Towards greater integration: Prospects for the development of agri-food trade between the EU and RCEP countries

Joanna Łukasiewicz[1], Bartłomiej Bajan[1,2]*, Karolina Pawlak[1]

1 Faculty of Economics, Department of Economics and Economic Policy in Agribusiness, Poznan University of Life Sciences, Poznań, Poland, 2 Faculty of Economics and Management, Department of Economics, Czech University of Life Sciences Prague, Prague, Czech Republic

* bartlomiej.bajan@up.poznan.pl

**Citation:** Łukasiewicz J, Bajan B, Pawlak K (2025) Towards greater integration: Prospects for the development of agri-food trade between the EU and RCEP countries. PLoS One 20(7): e0328866. https://doi.org/10.1371/journal.pone.0328866

## Abstract

The Regional Comprehensive Economic Partnership (RCEP) is the largest free trade agreement in the world, making it a potentially attractive trading partner for the European Union (EU). This paper aims to fill a research gap by examining agri-food trade between the EU and RCEP, with a focus on uncovering opportunities for sectoral cooperation and identifying competitive dynamics. It employs trade structure similarity and intra-industry trade analysis, including horizontal and vertical trade flows. Using the Export Similarity Index (ESI) and Product Similarity Index (PSI), the findings indicate stable, moderate competition in agri-food trade between the EU and RCEP, with similarity indices ranging from 0.32 to 0.35. While intra-industry trade remains limited due to geographic distance, there is notable potential for expanding high-quality EU agri-food exports. The results suggest that targeted trade policies could enhance cooperation by leveraging the EU's export strengths to meet RCEP demand. The study provides a strategic framework for future negotiations, emphasizing sectoral approaches to optimize trade outcomes and maximize trade benefits between the EU and RCEP countries.

## Introduction

The Regional Comprehensive Economic Partnership (RCEP) is the world's largest free trade agreement (FTA) to date, signed by the members of the Association of Southeast Asian Nations (ASEAN) and five ASEAN Dialogue Partners (i.e., Australia, China, Japan, New Zealand, and South Korea) [1]. The RCEP members represent around 30 percent of the world's population and global GDP and around one-third of international trade value [2]. The emergence of RCEP is part of the global trends observed in the so-called third wave of regionalism (following the Uruguay Round of WTO negotiations), which includes the emergence of mega-regional preferential trade agreements, which assume deep integration between countries or regions with

**Data availability statement:** All relevant data are within the manuscript and its Supporting Information files.

**Funding:** This study was founded by the National Science Centre, Poland, under the grant number 2022/47/O/HS4/00548. The funders had no role in study design, data collection and analysis, decision to publish, or preparation of the manuscript.

**Competing interests:** The authors have declared that no competing interests exist.

major shares in global trade [3]. The economic potential of RCEP and the substantive scope of the agreement can influence the intensification of economic and trade cooperation beyond the Indo-Pacific region [4]. In this context, we want to explore the potential trade strategy of the European Union (EU) applied to the RCEP. Specifically, we are interested in trade in agri-food products as one of the most vulnerable and socially discussed commodities.

On the one hand, as a high-demand market, the EU can play a pivotal role in global relations during the harmful uncertainty of the trade war between the US and China [5,6]. On the other hand, the EU failed negotiations of the Transatlantic Trade and Investment Partnership (TTIP) with the US [7] and deteriorated relations with China during the recent global pandemic [8] and the Russian-Ukrainian war [9]. However, in the case of the Asian direction, there still seems to be political will for agreement reflected, among others, in the Chinese Belt and Road Initiative (BRI) aimed at the economic consolidation of Eurasia [10] and many bilateral negotiations of trade agreements between the EU and Indo-Pacific countries.

The EU has already established significant trade agreements with several RCEP members, including South Korea (in force since 2015), Singapore (2019), Vietnam (2020), and New Zealand (2024), and is actively negotiating agreements with Indonesia, the Philippines, and Thailand [11]. Moreover, the Economic Partnership Agreement between the EU and Japan has been in force since February 2019 [11]. As the EU continues to deepen its trade partnerships in Asia, a mutual interest in trade liberalization between the EU and RCEP is evident. This shared goal underscores opportunities for trade growth and mutual economic benefits, even as both blocs navigate the challenges of overlapping trade agreements and the rise of regional mega-FTAs.

RCEP's implementation has already begun to reshape global trade flows. Pan et al. [12] show that while RCEP increases economic integration among its members, its effects are uneven, with Japan benefiting the most, while China's export scale is adversely affected by competing geopolitical strategies. Such dynamics pose new challenges for the European Union, which has pursued FTAs with several Indo-Pacific nations to counterbalance RCEP's influence and maintain its market access [13]. Breuss [14] highlights that RCEP fosters deepened intra-regional integration, with significant gains for its members, while creating trade diversion risks for non-members such as the EU. The author further suggests that overlapping FTAs may present unique opportunities for the EU to leverage its partnerships in Asia while navigating the complexities of mega-regional agreements like RCEP. More complexity is added by the geopolitical landscape of the US-China trade war. The U.S.-led Indo-Pacific Strategy and China's BRI create overlapping influences that reshape regional trade dynamics [15]. For the EU, these developments pose a dual challenge: leveraging its bilateral agreements with RCEP members to secure market access while navigating the effects of large-scale initiatives of the US and China.

The EU's recent experience negotiating mega-regional preferential trade agreements shows that a strong point of contention is issues related to agri-food trade. Such issues concerned or still concern negotiations of TTIP [16], the Comprehensive Economic and Trade Agreement (CETA) with Canada [17], and the EU-Mercosur

agreement [18]. In connection with the above negotiations, public concerns in EU countries have intensified, and there are doubts about the proper adjustment of food standards in negotiated agreements and, therefore, concerns about food safety [19].

The critical role of provisions regarding agri-food products in the ongoing and future trade negotiations raises the need to examine the sectoral implications of trade agreements. While the general evidence describing the impact of the RCEP on the EU exists, it does not provide sectoral-level analysis concerning agri-food products [20,21]. In the above-mentioned context, our study aims to examine agri-food trade between the EU and RCEP, focusing on uncovering opportunities for sectoral cooperation and identifying competitive dynamics. Thus, the novelty of our research lies in a detailed analysis of the agri-food trade between the EU and RCEP, covering 24 chapters on agri-food products. The research so far is concerned solely with macro-level relations between two groupings or relations between the EU and single countries in the Asia-Pacific region. Sectoral-level studies, to date, are of a descriptive nature or cover only a few groups of products; in contrast, we aim to analyze trade potential for each product category between the EU and RCEP based on empirical data. Some relevant policy assessments–Sustainability Impact Assessments–of ongoing negotiations exist [22]; however, these studies are limited to bilateral contexts. We aim to map a comprehensive view of the potential of agri-food trade between the EU and RCEP groupings.

Such a goal is reached firstly through analyzing export similarity in the world market between the EU and RCEP and secondly through analyzing changes in the intra-industry trade in agri-food products. Therefore, to the best of our knowledge, our study is the first attempt to map the baseline situation for potential negotiations of FTA between the EU and RCEP regarding agri-food products.

## Literature review

Since the end of the Uruguay Round of the GATT/WTO, preferential trade agreements have had an increasingly interregional dimension, aiming at the 'deep and comprehensive' liberalization of the goods, services, and investments market. Moreover, they improve conditions for cross-border business by reducing trade costs and eliminating regulatory barriers between the parties of the agreement [23]. This deep and comprehensive integration between countries or regions with significant shares in global trade and foreign direct investment contributes to the emergence of mega-regional preferential trade agreements [3], which can potentially influence trade rules and flows inside and outside their application area [24]. A vivid example of an active mega-regional trade agreement is the Regional Comprehensive Economic Partnership (RCEP), concluded in 2020. On the one hand, it is an important trading partner. On the other hand, it is a competitor to the EU economy, striving to maintain its current strong position in international trade [21].

The Regional Comprehensive Economic Partnership (RCEP) has the potential to significantly impact trade and economic cooperation not only in the Asia-Pacific area but also far beyond due to the agreement's substantive scope [4,25] and member countries' economic potential [26]. Kimura et al. [27] emphasize that the RCEP is essential for managing the worldwide uncertainty resulting from the ongoing war between Russia and Ukraine and the post-pandemic recovery of East Asia and ASEAN. It includes components essential to regional transformation, e.g., the first China-Japan-Korea free trade area. Such an agreement is essential for sustainable and inclusive growth and might enhance GVCs in the RCEP member countries.

The simulation studies of the possible impacts of the Regional Comprehensive Economic Partnership (RCEP) on the economies of the RCEP Member States and third countries reveal potential winners and losers. Petri and Plummer [28] assessed that by 2030, trade liberalization will have benefited the RCEP member countries to at least USD 428 billion, while trade diversion costs non-RCEP nations USD 48 billion. This is in line with the study by Estrades et al. [29], who also predicted a significant increase in intra-RCEP trade and a decline in trade between RCEP member countries and the rest of the world resulting from tariff and trade cost reductions within the RCEP. As suggested by the latter authors, only a productivity kick accompanying regional trade liberalization might have increased exports to the rest of the world. Estrades

et al. [29] showed that all participating countries can benefit from establishing a free trade area, although the gains are not distributed equally. Vietnam, Japan, and Cambodia are the nations with the most significant increases in exports, while Vietnam, the Philippines, and Japan see the most significant increases in imports. When considering the overall impact of the RCEP on exports and GDP, the Philippines, Vietnam, and Korea appear as the top gainers [30].

Moreover, the results of Suvannaphakdy [31] show that the trade effects of the RCEP vary across countries. According to their calculations, intra-ASEAN trade preferences will be undermined by tariff cuts under the RCEP, which will reallocate import sources from ASEAN members to more efficient RCEP partners. It is possible that Brunei, Indonesia, Laos, Myanmar, Malaysia, Singapore, and Thailand will experience relatively large export losses. At the same time, Cambodia, the Philippines, and Vietnam will likely suffer minimal losses from their exports because they sell mostly to the US, which is not a party to the RCEP. According to Zhou [32], increasing bilateral trade between RCEP members depends heavily on the economic scale. This is consistent with the gravity theory, which shows that economies with larger sizes are more capable of trade. The size of the population and geographic distance, which significantly impact trading logistics, are also important for shaping bilateral trade flows.

Scholars indicate that the RCEP will not substantially increase intra-regional trade in the short run since it is mainly a revised version of the previous ASEAN+1 free trade agreements [33]. On the other hand, it is worth emphasizing that pre-RCEP trade agreements have yet to result in significant tariff reductions in some cases. Therefore, the most significant increases in bilateral trade are expected between these pairs of countries that experienced the most profound tariff liberalization under the RCEP. China, Japan, and South Korea are among the possible greatest beneficiaries of the RCEP trade liberalization. Flach et al. [34] address this regularity, consistent with the customs union theory. When evaluating tariff efficiency under the RCEP, it can also be mentioned that the RCEP tariff regime is not highly preferential for most member states compared to the pre-existing ASEAN trade agreements. The study by Banh et al. [35] shows that the RCEP should deepen its provisions to better serve the regional trade integration goal.

Considering the world trade, an extensive cross-country analysis by Mahadevan and Nugroho [36] revealed the trade creation effect in all RCEP countries, except Singapore, along with a concurrent decline in the trade turnover of other Asian nations, the USA, and the EU. According to their estimates, there will probably be a slight decline in the overall value of EU trade due to the formation of the RCEP. However, due to the creation and diversion of trade, trade may fall in bilateral EU-RCEP relations. Flach et al. [34] also mention the anticipated shifts in trade structures brought about by the RCEP's trade creation and diversion effects, while Banga et al. [37] analyze their impacts on trade balances in individual member states. These findings highlight the potential benefits and drawbacks of the RCEP for different countries, creating the need for a nuanced understanding of its implications.

Another essential issue is that most of the above analyses refer to total trade flows, excluding a sectoral-level analysis. The same applies to more descriptive studies on the implications of signing the RCEP for the EU by Francois and Elsig [20], Hilpert [21], and Stehrer and Vujanovic [38]. Estrades et al. [29] presented one of the sectoral-specific approaches. They proved that meat products and food and beverages could be among the products subjected to a significant increase in exports from the RCEP. This is linked to reducing both tariffs and non-tariff barriers to trade. Considering more detailed analyses, Tong et al. [39] found the growth of China's fruit and vegetable exports to RCEP countries, resulting, among others, from the conclusion of the free trade area. As the RCEP formation can make smaller member countries lose their competitive advantages in the regional market, the advantageous trading destinations and export specializations in fish and seafood trade were identified for all RCEP economies by Erokhin et al. [40]. They noticed that China, Japan, and South Korea made good use of their potential and advantages in intraregional trade. However, none of the above studies covers a sectoral-level analysis of the RCEP agri-food trade with third countries, including the EU.

This issue deserves attention in light of the above-mentioned trade creation and diversion mechanisms resulting from regional integration and customs union theories. Those effects make it essential to identify the beneficial export specializations in all parties of the agreement to optimize the commodity structure of mutual trade and gains from trade

liberalization. For maximization of trade effects in all sectors, including agri-food trade, it is necessary to focus on the most competitive and, at the same time, demanded products in the trading partners' markets. Focusing on these product categories will contribute to the development of mutual trade and improve trade balance, one of the most simple and commonly accepted measures of international competitiveness at the sectoral level [41]. Being aware of the possible trade diversion effect in EU-RCEP relations, developing agri-food trade in line with such theories as Ricardo's theory of comparative advantages, Linder's theory of overlapping demand, and the Grubel-Lloyd theory of intra-industry trade makes it possible to take advantage from bilateral trade under the RCEP agreement. Finding key product groups with the most significant potential to compete on individual markets successfully and designing tailored domestic agricultural policies that enhance the chosen production and export specializations seem crucial for the EU-RCEP agri-food trade development. This paper offers an overview of the agri-food trade performance and competitive position of the EU and RCEP countries in bilateral trade by the Harmonized Commodity Description and Coding System (HS) chapters. Our detailed empirical analysis opens the floor for discussing further trade cooperation in the changing international environment due to institutional factors.

Rethinking the trade strategy that EU nations have been employing on the Asian and Pacific markets appears vital in light of the RCEP countries' growing geopolitical and economic potential in the global economy. In this context, our study attempts to identify the level of competition in agri-food trade between the EU and the RCEP and areas where, in the current baseline, it would be most optimal to strengthen trade cooperation. Our research contributes threefold to the existing literature on trade cooperation between the EU and RCEP countries. Firstly, we refer to trade in agri-food products as being one of the most sensitive when negotiating trade liberalization at a regional scale while at the same time excluded from previous studies. Secondly, we analyze both the similarity of agri-food exports between the EU and RCEP and the intensity and structure of intra-industry trade in agri-food products using disaggregated trade data in the long run, which makes our analysis reliable, well matching the reality and showing changes over time. Thirdly, we offer the baseline for further trade developments between the EU and RCEP, including those within the possible free trade area. To the best of our knowledge, our study is the first attempt to map the baseline conditions for potential negotiations of the preferential trade agreement between the EU and RCEP regarding agri-food products.

## Materials and methods

### Data collection

The study was conducted for the first 24 chapters classified in the HS, which includes the agri-food trade. The full nomenclature of the groups of agri-food products is included in the supporting information (S1 Appendix). In order to eliminate one-year fluctuations, the study was conducted for five two-year periods: 2013/2014, 2015/2016, 2017/2018, 2019/2020 and 2021/2022.

Firstly, we analyze the similarity of agri-food exports between the EU (27 members as of 1st of May 2024) and RCEP (14 members as of 1st of May 2024) using two indicators, namely the Export Similarity Index (ESI) and the Product Similarity Index (PSI). The idea behind these calculations comes from the literature indicating that higher similarity of exports in a given market shows greater competition among exporters [42]. Therefore, we are able to identify changes in the potential competition in agri-food trade between the RCEP and the EU. The calculations were made using World Integrated Trade Solution (WITS) [43] 6-digit level data, giving 722 agri-food products. We aimed for the highest possible level of disaggregation as it significantly impacts the values of the indicators we used [44]. As the level of data disaggregation increases, their values tend to decrease, giving a more accurate estimation [45]. We used export data at the world market to calculate both indicators, subtracting trade values between countries inside the RCEP and the EU, respectively.

In the next step, the intensity and nature of intra-industry trade between EU and RCEP countries were calculated. For this purpose, data from the Comext-Eurostat database were used. To accurately analyze intra-industry trade, calculations were made at the level of 8-digit Combined Nomenclature (CN) aggregation, giving 4689 agri-food products. Using

highly disaggregated data can help conduct an in-depth analysis and obtain more information on the import and export patterns of the countries, which allows a better understanding of trade dynamics [46]. Disaggregated data enables the study to identify trade patterns specific to individual product categories, leading to more accurate measurements of trade similarity and intra-industry trade levels. By tracking exports and imports at the 6-digit HS and 8-digit CN code level, the analysis can capture variations across a broad range of agri-food products, enhancing the specificity and relevance of the results [47].

## Calculation Procedure

The ESI, originally used by Finger & Kreinin [48], is a simple measure of structure similarity, aiming at the comparison of export patterns. Its values range from 0 to 1, where 0 means a complete lack of similarity and 1 indicates identical trade structures. ESI can be written as:

$$ESI_{ij,d} = \sum_{c=1}^{n} \min \left( x_{i,d}^{c}, x_{j,d}^{c} \right)$$

(1)

where: $x$ indicates shares of exported goods in the total agri-food export, $c$ denotes exported commodity, $i$ and $j$ represent trading competitors (in our case, RCEP and EU, respectively), and $d$ indicates the targeted market where the products are exported.

The PSI indicates how much the absolute values of export streams overlap, going beyond their structures. Higher PSI values suggest greater similarity in exports, which, in turn, implies increased competitive pressure between compared partners [49]. The extreme values of the PSI are interpreted similarly to the extreme values of the ESI. The PSI can be written as:

$$PSI = 1 - \left[ \frac{\sum_{c=1}^{n} \left| x_{i,d}^{c} - x_{j,d}^{c} \right|}{\sum_{c=1}^{n} \left( x_{i,d}^{c} + x_{j,d}^{c} \right)} \right]$$

(2)

where: all the variables are as previously defined.

The ESI and PSI indices were chosen as primary tools for analyzing export structure because they enable an assessment of the degree of export similarity between the EU and RCEP, which is essential for determining the level of competition in the global agri-food market. In the literature, these indices are frequently used to analyze structural similarities in international trade, making them suitable tools for assessing the level of competition [49]

First, to assess the strength of intra-industry trade, the classic Grubel-Lloyd index of intra-industry trade was calculated [50]:

$$IIT_{ik} = 1 - \frac{\left| X_{ik} - M_{ik} \right|}{\left( X_{ik} + M_{ik} \right)}$$

(3)

where: $X_{ik}$ and $M_{ik}$ denote export and import of a given category of products $i$ from/to country $k$ (overall or bilateral). The $IIT$ index assumes values between 0 and 1. The higher values on this index suggest a greater overlap between a country's export and import flows, indicating a higher degree of intra-industry trade [51]. According to Qasmi & Fausti [52], four classes of intra-industry trade intensity can be identified when evaluating the performance of the $IIT$ index:

1) $0.00 < IIT \le 0.25$ – strong inter-industry trade;

2) $0.25 < IIT \le 0.50$ – weak inter-industry trade;

3) $0.50 < IIT \leq 0.75$ – weak intra-industry trade;

4) $0.75 < IIT \leq 1.00$ – strong intra-industry trade.

The Grubel-Lloyd assesses trade values in single years, making it a static measure suitable for assessing patterns of specialization (inter- or intra-industry) at a specific point in time. This indicator does not show whether there are any long-term adjustments in trade patterns. To remove this limitation, the study also includes an analysis of marginal intra-industry trade (*MIIT*), a dynamic measure developed by Hamilton & Kniest [53]. The *MIIT* index is calculated according to the formula:

$$MIIT_{ik} = 1 - \frac{|\Delta X_{ik} - \Delta M_{ik}|}{|\Delta X_{ik}| + |\Delta M_{ik}|}$$

(4)

where: $\Delta X_{ik}$ and $\Delta M_{ik}$ denote changes in export and import values in two investigated years or periods. Similarly to *IIT*, *MIIT* ranges from 0 to 1. Values close to these extremes correspond to the observed change in the character of trade towards strongly inter-industry (0) or intra-industry (1) trade, respectively [54]. This approach identifies how trade structures adjust over time, providing a more comprehensive picture of trade dynamics.

To analyze quality changes in trade by dividing intra-industry trade (*IIT*) into horizontal and vertical trade, a method using relative unit values (UVs) was used [55]. This method has been previously used in several analyses [56–58]. According to it, horizontal intra-industry trade (*HIIT*) is identified when the ratio of unit values of exports (*UVX*) to imports (*UVM*) is within the range:

$$1 - \alpha \leq \frac{UV_{ik}^X}{UV_{ik}^M} \leq 1 + \alpha$$

(5)

where: $\alpha$ is set at 0.15 as per Greenaway et al. (1994). When *IIT* values do not fit within this range, the trade is categorized as vertical intra-industry trade (*VIIT*). As defined by Greenaway et al. [55], the distinction between high- and low-quality *VIIT* is determined by the following conditions:

$$\frac{UV_{ik}^X}{UV_{ik}^M} > 1 + \alpha \quad \text{or} \quad \frac{UV_{ik}^X}{UV_{ik}^M} < 1 - \alpha$$

(6)

A ratio of export prices to import prices above 1.15 indicates high-quality vertical intra-industry trade (*VIIThigh*), meaning that a country (in the case of our study – the EU) exports higher-quality goods while importing lower-quality products. Conversely, a ratio below 0.85 indicates low-quality vertical intra-industry trade (*VIITlow*), where a country (the EU) exports lower-quality goods compared to imports. In the analysis of intra-industry trade, the CN classification at the 8-digit level was used, which is a more detailed extension of the HS classification. This allows each CN code to be assigned to the appropriate product group at the 2-digit HS level. Ratios of unit values of exports (*UVX*) to imports (*UVM*) are calculated at the level of 8-digit CN codes. Based on the results for individual CN codes, the value of products traded within horizontal and vertical trade can then be determined for each product group classified at the 2-digit HS level. This approach makes it possible to obtain aggregated indicators for each general product group at the 2-digit HS level. In other words, the analysis reveals the value of products within a given group (2-digit HS) that constitutes a specific type of intra-industry trade, based on classifications made at the detailed CN level.

The intra-industry trade (*IIT*) analysis, conducted using the classic Grubel-Lloyd index and the marginal *MIIT* index, will allow us to answer the research question concerning cooperation opportunities between the EU and RCEP, particularly

with a sectoral approach in mind. Following Greenaway et al.'s [55] approach, with low- and high-quality intra-industry trade values available, it is also possible to calculate the *Quality-Differentiated IIT* index that takes these values into account:

$$QD-IIT_{ik} = 1 - \left( \frac{\left|X_{ik}^{HIIT} - M_{ik}^{HIIT}\right| + \left|X_{ik}^{VIITlow} - M_{ik}^{VIITlow}\right| + \left|X_{ik}^{VIIThigh} - M_{ik}^{VIIThigh}\right|}{X_{ik} + M_{ik}} \right)$$

(7)

As in the case of the standard Grubel-Lloyd index, the index values range from 0 to 1, and the higher and closer to 1 the values, the more intensive intra-industry trade can be observed.

## Results and discussion

Results for the similarity of exports between EU and RCEP are very similar in each analyzed period for ESI and PSI, ranging from 0.32 to 0.35 (Fig 1). Such results indicate a relatively stable competitive situation in agri-food trade between the two groupings. The likelihood of the ESI and the PSI being similar or different depends mainly on several factors, such as economic scale [59] and product specialization [60]. In the case of the EU and the RCEP, the similarity between both indicators comes mainly from similar scales of operation in external agri-food exports.

The literature analyzing export similarity using the ESI and PSI indicators has accepted that there is no simple way to determine their specific values that can indicate similarity and, thus, the presence of competition [61]. An often-accepted,

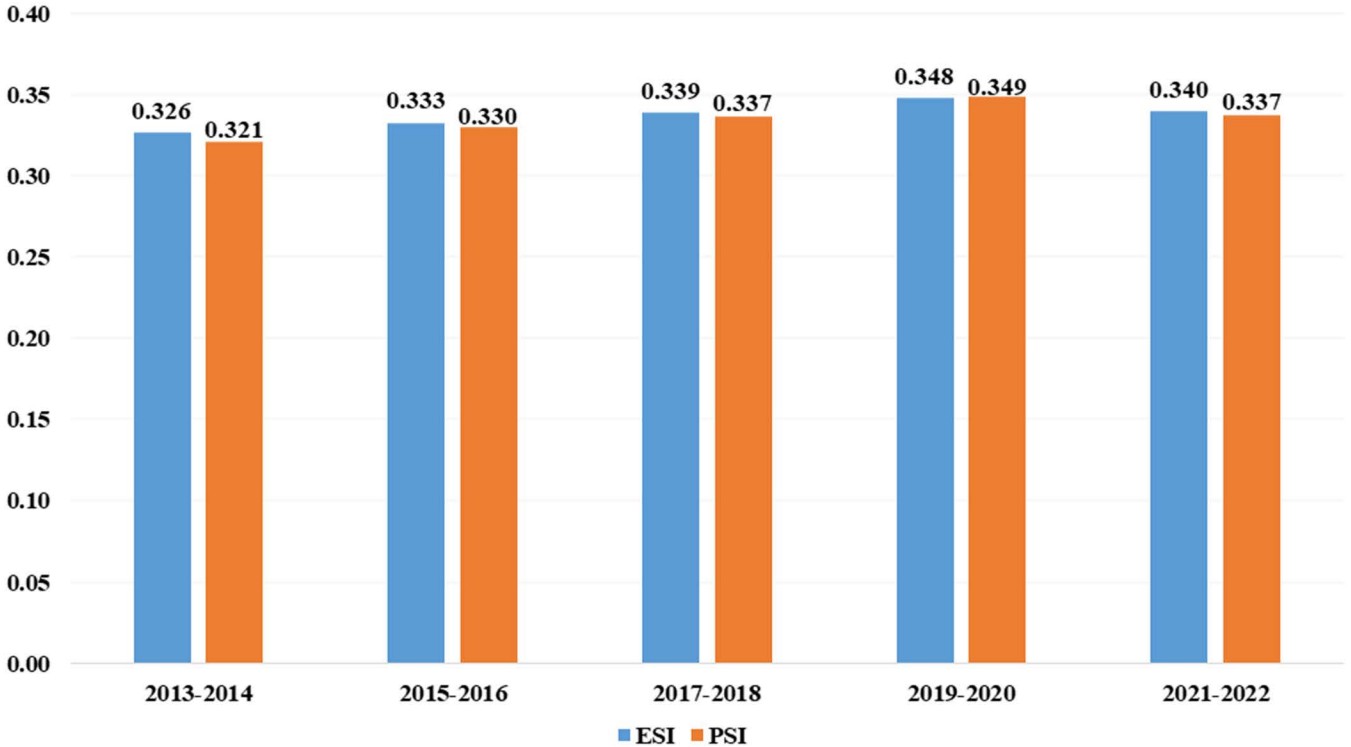

**Fig 1. ESI and PSI values between EU and RCEP considering external agri-food exports.** Source: Own calculations based on World Integrated Trade Solution (WITS) HS 6-digit data.

practical rule of thumb is to consider a value of 0.5 (50%) as indicative of similarity [62]. However, this solution can be significantly inaccurate due to differences in values depending on the level of data aggregation [44]. For this reason, it is recommended that the results be related to the broader context and that the presence of similarity be determined based on comparisons with results for other groupings or countries [49].

The best point of reference for us is the study by Antimiani et al. [62], where data by HS classification at the 6-digit level was used, as in our case. This study showed that PSI values for the Czech Republic, Hungary, and Poland with selected EU-15 countries (EU old Member States from at least 1995) were, on average, similar to our study only in trade to the EU-12 countries (which entered the EU after 2003) in 2006−2007, i.e., after joining the customs union. PSI values before accession and in other markets were significantly lower than in our study. Moreover, the export similarity results obtained by Rondinella et al. [42] for Mediterranean EU countries based on trade in the EU-25 market from 2005 to 2012 show lower similarity on average for PSI and similar results to our study for ESI. However, it should be noted that the authors' analysis was conducted for CN classification at the 8-digit level, reducing the values of the obtained indicators. Neverthe-less, the countries they studied show high similarity in agri-food trade from a product and volume point of view. Therefore, high ESI and PSI values are expected. Based on these comparisons, in our case, the ESI and PSI results indicate that the EU and RCEP have significant potential to compete in exporting agri-food products to the world market. In order to identify specific areas of agri-food trade in which the two groupings manifest the potential to strengthen trade cooperation, we developed an analysis of intra-industry trade.

In the case of the EU and RCEP countries, the value of their bilateral trade in agri-food products increased between 2013 and 2022. Agri-food exports to RCEP countries increased by about 95%, while imports increased by about 51% (S1 Fig). Considering the overall exchange of goods classified in the first 24 HS product groups, a significant overlap of export and import flows could be observed during the analyzed period. However, analyzing individual groups of agri-food products, strong intra-industry trade between EU and RCEP countries was not observed in any of them (Table 1, Fig 2). Non-tariff barriers heavily influence trade between these regions. In particular, our results were primarily affected by the considerable distance between EU and RCEP countries. Given the cost of transportation and the sensitivity of some agri-food commodities to transportation, the greater the distance between countries, the more apparent the negative impact on strengthening the intensity of intra-industry trade.

Such a conclusion is confirmed by Caetano and Galego [63], who, studying European countries, looked for the deter-minants of intra-industry trade and found that distance between countries is one of the most important of them. Distance between countries is also negatively correlated with the development of horizontal trade, i.e., products of similar quality (HIIT) [64]. Our study confirms such a finding since trade of goods of similar quality in almost all product groups had a negligible share. This was particularly evident with animal products (HS-05), dairy products (HS-04), fruits, and nuts (HS-08), where the maximum value of the HIIT index in analyzed years was 0.06 in the HS-08 group (S1 Table). These prod-ucts can be categorized as sensitive and difficult to transport over long distances. However, distance has also resulted in a low significance of horizontal trade in the groups of processed products. Such a situation was observed, for example, in the group of processed foods (HS-21), sugar and confectionery (HS-17), or tobacco (HS-24). In addition, sanitary and phytosanitary measures are highlighted as critical non-tariff barriers affecting trade in agricultural products between the EU and RCEP [65]. These measures are necessary to protect human, animal, or plant life but can often be trade barriers due to stringent requirements or lack of harmonization across regions.

Intra-industry trade could not be assessed as intensive from 2013 to 2022, and weak intra-industry trade ($0.50 < IIT \le 0.75$) was observed only in some groups of agri-food products, such as animal products (HS-05), live trees and other plants (HS-06), and the processed food group (HS-21). In the case of these product groups, weak intra-industry trade could be observed throughout the analyzed period, where low- and high-quality vertical trade flows dominated. In addition, increasingly intensive intra-industry trade between the EU and RCEP was observed from 2015/2016 in lac; gums, resins, and other vegetable saps and extracts (HS-13). High-quality vertical exchange (VIIThigh) in HS-13 goods

Table 1. Intra-industry trade and marginal intra-industry trade in agri-food products between the EU and RCEP countries by HS code (the EU perspective).

| HS¹ | 2013/2014 | | | | 2015/2016 | | | | 2017/2018 | | | | 2019/2020 | | | | 2021/2022 | | | | MIIT |
|---|---|---|---|---|---|---|---|---|---|---|---|---|---|---|---|---|---|---|---|---|---|
| | QD-IIT | HIIT | VIIT-low | VIIThigh | QD-IIT | HIIT | VIIT-low | VIIThigh | QD-IIT | HIIT | VIIT-low | VIIThigh | QD-IIT | HIIT | VIIT-low | VIIThigh | QD-IIT | HIIT | VIIT-low | VIIThigh | |
| 1 | 0.25 | 0.06 | 0.03 | 0.16 | 0.39 | 0.00 | 0.38 | 0.01 | 0.23 | 0.08 | 0.13 | 0.02 | 0.26 | 0.01 | 0.14 | 0.02 | 0.27 | 0.02 | 0.14 | 0.11 | 0.85 |
| 2 | 0.43 | 0.18 | 0.18 | 0.07 | 0.20 | 0.04 | 0.16 | 0.00 | 0.24 | 0.08 | 0.10 | 0.06 | 0.17 | 0.08 | 0.03 | 0.06 | 0.15 | 0.03 | 0.10 | 0.02 | 0.01 |
| 3 | 0.33 | 0.04 | 0.15 | 0.14 | 0.24 | 0.01 | 0.10 | 0.13 | 0.34 | 0.06 | 0.15 | 0.13 | 0.36 | 0.07 | 0.15 | 0.14 | 0.38 | 0.05 | 0.15 | 0.18 | 0.94 |
| 4 | 0.19 | 0.00 | 0.01 | 0.18 | 0.15 | 0.00 | 0.01 | 0.14 | 0.10 | 0.02 | 0.01 | 0.07 | 0.07 | 0.02 | 0.01 | 0.04 | 0.08 | 0.01 | 0.00 | 0.07 | 0.05 |
| 5 | 0.71 | 0.00 | 0.52 | 0.19 | 0.79 | 0.00 | 0.65 | 0.14 | 0.75 | 0.00 | 0.59 | 0.16 | 0.72 | 0.01 | 0.53 | 0.18 | 0.70 | 0.00 | 0.53 | 0.17 | 0.88 |
| 6 | 0.60 | 0.05 | 0.09 | 0.46 | 0.55 | 0.00 | 0.03 | 0.52 | 0.54 | 0.02 | 0.09 | 0.43 | 0.55 | 0.02 | 0.08 | 0.45 | 0.55 | 0.02 | 0.14 | 0.39 | 0.90 |
| 7 | 0.37 | 0.11 | 0.09 | 0.17 | 0.41 | 0.12 | 0.09 | 0.20 | 0.43 | 0.15 | 0.13 | 0.15 | 0.51 | 0.14 | 0.14 | 0.23 | 0.47 | 0.18 | 0.16 | 0.13 | 0.78 |
| 8 | 0.29 | 0.01 | 0.06 | 0.22 | 0.22 | 0.01 | 0.06 | 0.15 | 0.19 | 0.00 | 0.05 | 0.14 | 0.24 | 0.03 | 0.06 | 0.15 | 0.19 | 0.06 | 0.02 | 0.11 | 0.06 |
| 9 | 0.08 | 0.00 | 0.01 | 0.07 | 0.19 | 0.02 | 0.01 | 0.16 | 0.20 | 0.02 | 0.01 | 0.17 | 0.24 | 0.01 | 0.00 | 0.23 | 0.25 | 0.00 | 0.01 | 0.24 | 0.68 |
| 10 | 0.05 | 0.01 | 0.01 | 0.03 | 0.04 | 0.00 | 0.02 | 0.02 | 0.07 | 0.00 | 0.03 | 0.04 | 0.02 | 0.00 | 0.01 | 0.01 | 0.05 | 0.00 | 0.04 | 0.01 | 0.56 |
| 11 | 0.15 | 0.02 | 0.02 | 0.11 | 0.14 | 0.05 | 0.02 | 0.07 | 0.13 | 0.02 | 0.02 | 0.09 | 0.17 | 0.03 | 0.02 | 0.12 | 0.21 | 0.01 | 0.08 | 0.12 | 0.31 |
| 12 | 0.29 | 0.14 | 0.10 | 0.05 | 0.39 | 0.16 | 0.16 | 0.07 | 0.41 | 0.01 | 0.34 | 0.06 | 0.53 | 0.02 | 0.46 | 0.06 | 0.33 | 0.15 | 0.13 | 0.05 | 0.42 |
| 13 | 0.45 | 0.36 | 0.00 | 0.09 | 0.51 | 0.41 | 0.00 | 0.10 | 0.73 | 0.05 | 0.13 | 0.55 | 0.70 | 0.02 | 0.26 | 0.42 | 0.75 | 0.01 | 0.00 | 0.74 | 0.69 |
| 14 | 0.05 | 0.00 | 0.00 | 0.05 | 0.03 | 0.00 | 0.03 | 0.00 | 0.04 | 0.00 | 0.04 | 0.00 | 0.04 | 0.00 | 0.04 | 0.00 | 0.02 | 0.00 | 0.02 | 0.00 | 0.00 |
| 15 | 0.23 | 0.01 | 0.02 | 0.20 | 0.23 | 0.01 | 0.01 | 0.21 | 0.16 | 0.08 | 0.04 | 0.04 | 0.14 | 0.07 | 0.02 | 0.05 | 0.11 | 0.02 | 0.06 | 0.03 | 0.80 |
| 16 | 0.21 | 0.01 | 0.02 | 0.18 | 0.19 | 0.01 | 0.01 | 0.17 | 0.26 | 0.02 | 0.12 | 0.12 | 0.26 | 0.03 | 0.10 | 0.13 | 0.21 | 0.08 | 0.03 | 0.10 | 0.45 |
| 17 | 0.31 | 0.02 | 0.18 | 0.11 | 0.37 | 0.06 | 0.20 | 0.11 | 0.43 | 0.11 | 0.26 | 0.06 | 0.44 | 0.06 | 0.17 | 0.21 | 0.49 | 0.00 | 0.33 | 0.16 | 0.38 |
| 18 | 0.41 | 0.01 | 0.00 | 0.40 | 0.52 | 0.41 | 0.01 | 0.10 | 0.36 | 0.02 | 0.02 | 0.32 | 0.44 | 0.38 | 0.01 | 0.05 | 0.09 | 0.03 | 0.02 | 0.04 | 0.08 |
| 19 | 0.29 | 0.02 | 0.05 | 0.22 | 0.27 | 0.02 | 0.06 | 0.19 | 0.22 | 0.02 | 0.15 | 0.05 | 0.33 | 0.01 | 0.15 | 0.17 | 0.31 | 0.05 | 0.09 | 0.17 | 0.39 |
| 20 | 0.60 | 0.25 | 0.03 | 0.32 | 0.38 | 0.13 | 0.06 | 0.19 | 0.50 | 0.15 | 0.11 | 0.24 | 0.52 | 0.08 | 0.11 | 0.33 | 0.58 | 0.08 | 0.14 | 0.36 | 0.79 |
| 21 | 0.53 | 0.00 | 0.00 | 0.53 | 0.61 | 0.00 | 0.02 | 0.59 | 0.62 | 0.00 | 0.03 | 0.59 | 0.62 | 0.07 | 0.03 | 0.52 | 0.62 | 0.10 | 0.04 | 0.48 | 0.96 |
| 22 | 0.15 | 0.01 | 0.04 | 0.10 | 0.15 | 0.04 | 0.04 | 0.07 | 0.15 | 0.02 | 0.03 | 0.10 | 0.17 | 0.02 | 0.04 | 0.11 | 0.18 | 0.02 | 0.04 | 0.12 | 0.27 |
| 23 | 0.35 | 0.06 | 0.26 | 0.03 | 0.39 | 0.05 | 0.30 | 0.04 | 0.61 | 0.17 | 0.39 | 0.05 | 0.65 | 0.03 | 0.39 | 0.20 | 0.58 | 0.09 | 0.45 | 0.04 | 0.72 |
| 24 | 0.55 | 0.02 | 0.13 | 0.40 | 0.66 | 0.00 | 0.19 | 0.47 | 0.37 | 0.00 | 0.10 | 0.27 | 0.28 | 0.00 | 0.08 | 0.20 | 0.31 | 0.00 | 0.09 | 0.23 | 0.08 |

¹Full nomenclature of the analyzed product groups: 1 – Live animals; animal products; 2 – Meat and edible meat offal; 3 – Fish and crustaceans, molluscs and other acquatic invertebrates; 4 – Dairy produce; birds' eggs; natural honey; edible products of animal origin, not elsewhere specified or included; 5 – Products of animal origin, not elsewhere specified or included; 6 – Live trees and other plants; bulbs, roots and the like; cut flowers and ornamental foliage; 7 – Edible vegetables and certain roots and tubers; 8 – Edible fruit and nuts; peel of citrus fruit or melons; 9 – Coffee, tea, mate and spices; 10 – Cereals; 11 – Products of the milling industry; malt; starches; inulin; wheat gluten; 12 – Oil seeds and oleaginous fruits; miscellaneous grains, seeds and fruit; industrial or medicinal plants; straw and fodder; 13 – Lac; gums, resins and other vegetable saps and extracts; 14 – Vegetable plaiting materials; vegetable products not elsewhere specified or included; 15 – Animal or vegetable fats and oils and their cleavage products; prepared edible fats; animal or vegetable waxes; 16 – Preparations of meat, of fish or of crustaceans, molluscs or other aquatic invertebrates; 17 – Sugars and sugar confectionery; 18 – Cocoa and cocoa preparations; 19 – Preparations of cereals, flour, starch or milk; pastrycooks' products; 20 – Preparations of vegetables, fruit, nuts or other parts of plants; 21 – Miscellaneous edible preparations; 22 – Beverages, spirits and vinegar; 23 – Residues and waste from the food industries; prepared animal fodder; 24 – Tobacco and manufactured tobacco substitutes.

Source: own calculations based on Comext-Eurostat, CN 8-digit data.

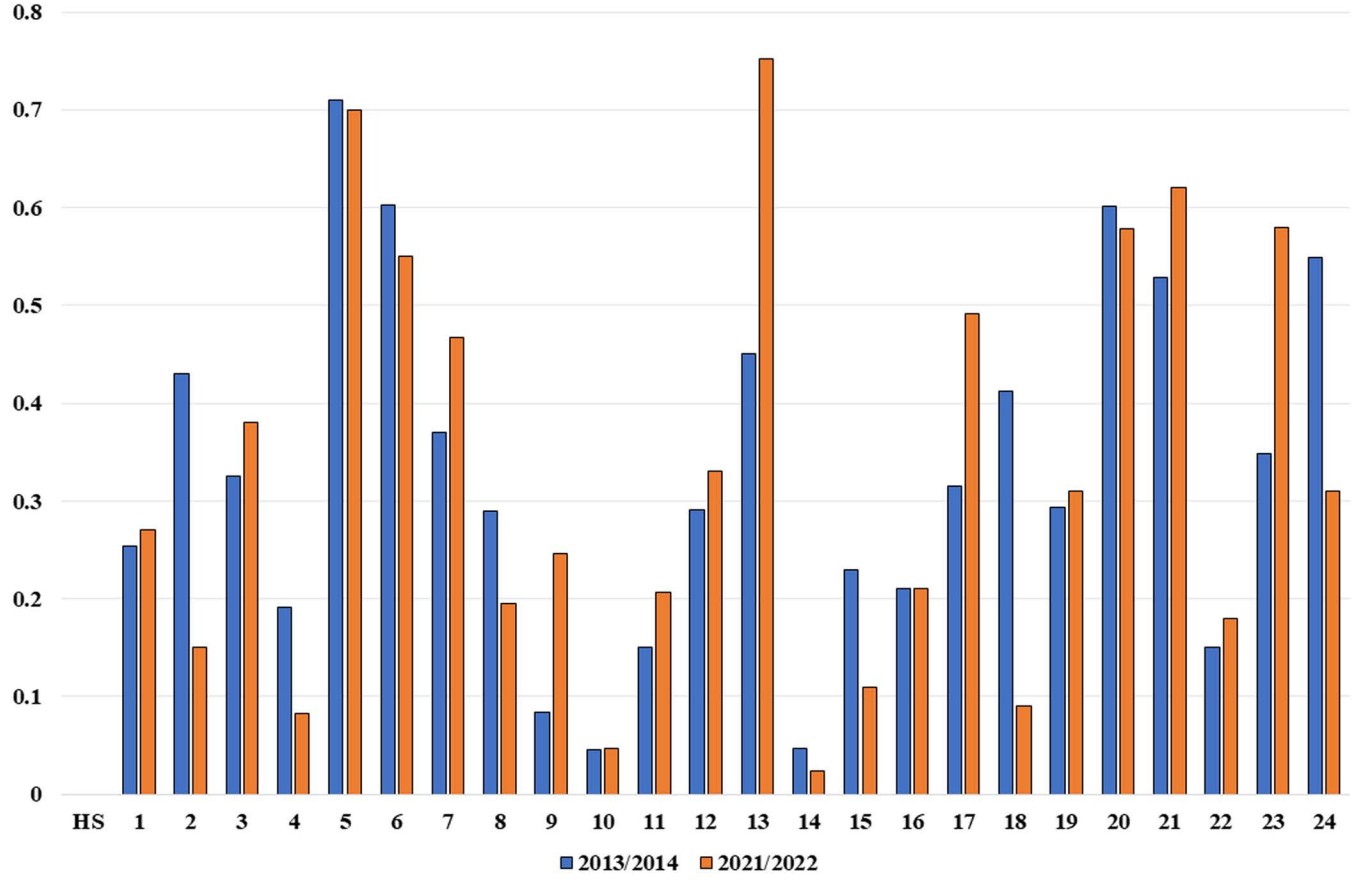

**Fig. 2. Changes in intra-industry trade in agri-food products between 2013/2014 and 2021/2022.** Source: Own calculations based on Comext-Eurostat, CN 8-digit data.

strengthened from period to period. In 2021/2022, 69% of the products in this group qualified for high-quality vertical trade, and their value accounted for 91% of intra-industry trade within the HS-13 group. A similar situation and increasing share of high-quality trade was observed for product groups such as preparations of vegetables, fruit, nuts or other parts of plants (HS-20) and miscellaneous edible preparations (HS-21), indicating that RCEP countries received higher quality goods than exported goods. This is also confirmed by Kostadinov [66], who examined the prospects for food trade between the EU and China. According to the author, trade between China and the EU has grown significantly in recent years, as rising incomes and a significant population in China spur demand for high-quality food. Such results also indicate that EU countries specialize in exporting higher value-added goods [67].

An important policy tool for enhancing high-quality EU agri-food exports lies in promoting and protecting Geographical Indications (GIs). These region-specific labels safeguard products with unique qualities linked to their origin and production practices. Empirical studies show that GIs often command significantly higher unit values in international markets, providing a clear competitive edge for exporters [68,69]. Moreover, GIs play a prominent role in EU trade negotiations, frequently appearing as strategic provisions in FTAs, including those with RCEP partners such as Japan (over 200 European Geographical Indications were recognized and protected) [70] and South Korea (dedicated protocol on GIs is protecting around 60 EU names) [71]. As highlighted by Belletti et al. [68], GIs improve market access and contribute to rural development and producer incomes, supporting the sustainability dimension of EU trade policy. In the context of EU–RCEP

relations, extending GI protection—particularly in countries without bilateral FTAs—could enhance high-value exports by removing imitations, aligning standards, and increasing consumer trust. This would directly support the observed high-quality vertical intra-industry trade (VIIThigh) flows in key agri-food sectors.

Existing policy-level assessments corroborate our findings. For instance, the EU Sustainability Impact Assessment of the EU–Indonesia Free Trade Agreement predicts that the agreement could lead to increased production of dairy and alcoholic beverages in the EU, driven by enhanced market access and trade liberalization [72]. This aligns with our empirical findings, where *VIIThigh* trade patterns are observed in categories such as beverages, dairy, and prepared foods. The consistency between EU economic modeling and our trade analysis supports the view that targeted trade negotiations, including GI recognition and product-specific liberalization, could significantly strengthen the EU's competitive position in Indo-Pacific markets.

Estrades et al. [29] found that agricultural exports from RCEP countries will grow significantly by 2035. Among the sectors experiencing the fastest export growth rates are meat products and food and beverages. Such findings correspond to our results in intra-industry trade between the EU and RCEP countries, where meat products (HS-01, HS-05) are characterized by significant levels of the *MIIT* index, showing trade transformations strengthening intra-industry trade in the analyzed period. Moreover, changes toward strengthening intra-industry trade were observed with products such as miscellaneous edible preparations, fish and crustaceans, molluscs and other aquatic invertebrates, and live trees and other plants. However, the analysis of all agri-food groups is a novelty, so we did not find many studies similar to ours. Usually, studies focus on the analysis of single sectors using a macroeconomic approach to analyze the trade of RCEP countries, e.g., [29,65,73], or descriptive studies are performed, e.g., [74,75]. Therefore, our study contributes to filling a knowledge gap in the domain.

Economic integration seems to be a key factor supporting processes related to the development of trade. For instance, Jiang [76], analyzing the situation in China using the Grubel-Lloyd index, noted strong and increasing trends in intra-industry trade with most RCEP countries between 2001 and 2021. Moreover, due to the existence of ASEAN or RCEP, an increase in the importance of intra-regional trade in Asian countries can be observed [77,78]. In contrast, a study by Chen and Lombaerde [79] found that the share of intra-regional trade for RCEP countries in 2010 was 41.2%, while for EU countries, it was 62.2%. In 2018, it remained at a similar level, with the share of intra-regional trade at around 40% for RCEP countries and around 58% for EU countries. In the case of trade between the analyzed groupings, it amounted to more than 10% of their total exports [80]. Therefore, strengthening trade cooperation could be mutually beneficial, increasing the competitiveness of both regions in the global market, which would promote economic growth and allow the development of new markets for products that were restricted by tariff barriers, improving the availability and variety of goods for consumers.

## Conclusions

Agri-food trade between the EU and RCEP shows the potential for competition and, at the same time, the potential for cooperation. The values of the export similarity index (ESI) and product similarity index (PSI) indicate stable, moderate competition between the EU and RCEP countries in the agri-food sector on the global market. However, taking into account the differences in the quality of the agri-food products traded, and thus the differences in the specialization of analyzed groupings, in addition to the competitive potential, a promising direction for enhancing trade cooperation is observed.

In the case of intra-industry trade in agri-food commodities between the EU and RCEP, high-quality vertical trade was observed in most product groups, indicating the higher quality of products exported from the EU. Among all products where intra-industry trade occurred, there was a high-quality vertical trade for 49% of them, 15% was horizontal trade and 36% was low-quality vertical trade. Despite the lack of intensive intra-industry trade between the EU and RCEP in the analyzed period, strengthening weak intra-industry trade was observed. Moreover, high MIIT values heading towards

one were observed in many agri-food product groups, indicating the development of both imports and exports in EU and RCEP countries and the strengthening of intra-industry trade, which is a positive aspect in light of possible future trade cooperation.

Therefore, key findings include moderate competition in agri-food trade between the EU and RCEP and the dominance of high-quality vertical trade flows (*VIIThigh*), suggesting the EU's competitive advantage in high-value agri-food products. We also observed weak but strengthening intra-industry trade in select product groups, highlighting untapped potential for trade cooperation. These insights underscore the need for harmonized strategic export policies to develop trade relations between EU and RCEP countries.

In enhancing trade cooperation between the EU and RCEP, it would be essential to apply targeted trade policies to specific branches of the agri-food sector that would consider the competitive context and intra-industry trade dynamics. Such policies could consider specific conditions in both markets and leverage the groupings' strengths from the perspective of the quality of agri-food products. We recommend setting up bilateral working groups on the harmonization of sanitary and phytosanitary standards to compare and align standards between the EU and RCEP, such as permissible levels of pesticide residues in agricultural products. In addition, we propose the creation of joint certification centers, which could be set up in RCEP countries (which are part of a trade agreement with the EU) to speed up export processes. This approach would reduce the number of rejections of goods at borders due to failure to meet standards and decrease the time and cost of bringing products into compliance with EU requirements. Implementing these recommendations would require coordination among various market actors, including governments, international organizations, and the private sector, to effectively adapt to the dynamically changing global trade environment. However, it could bring many mutual economic benefits and positively impact the diversity of goods in both markets.

This study contributes to the theoretical discourse on regional trade integration by illustrating how structural trade differences in agri-food products affect intra-industry trade by empirically linking quality differentiation (e.g., *VIIThigh* and *VIITlow*) to trade potential. Our study maps the current landscape of EU-RCEP agri-food trade and provides a strategic framework for enhancing mutual trade benefits. By focusing on competitive strengths and cooperative opportunities, the EU can effectively navigate the complex trade environment posed by the RCEP agreement. Conversely, a limitation of our study is the failure to measure a specific effect of non-tariff barriers, such as sanitary and phytosanitary regulations and environmental regulations. Restrictive EU standards for food quality, health protection, and $CO_2$ emissions do not always align with those of RCEP countries, resulting in the need for additional certification and product adjustments, which can hinder trade.

Further research is recommended to explore the long-term effects of the EU's ongoing and prospective FTAs with RCEP countries. Given the significant exports and imports of agri-food products in the EU and RCEP countries, it would be essential to examine how strongly the effects of trade shifting and creation could be observed due to intensifying trade cooperation between these groupings. Future studies should also consider non-tariff barriers and their impact on trade. For this purpose, it would be possible to use regression analysis or a gravity model.

## Supporting information

**S1 Appendix. Full nomenclature of the analyzed product groups.**
(DOCX)

**S1 Fig. Value of agri-food trade between the EU and RCEP in 2013–2022 (billions EUR).**
(DOCX)

**S1 Table. Descriptive statistics for intra-industry trade.**
(DOCX)

## Author contributions

**Conceptualization:** Joanna Łukasiewicz, Bartłomiej Bajan, Karolina Pawlak.

**Data curation:** Joanna Łukasiewicz, Bartłomiej Bajan.

**Formal analysis:** Bartłomiej Bajan.

**Funding acquisition:** Karolina Pawlak.

**Investigation:** Joanna Łukasiewicz.

**Methodology:** Joanna Łukasiewicz, Bartłomiej Bajan.

**Project administration:** Karolina Pawlak.

**Supervision:** Karolina Pawlak.

**Visualization:** Bartłomiej Bajan.

**Writing – original draft:** Joanna Łukasiewicz, Bartłomiej Bajan, Karolina Pawlak.

**Writing – review & editing:** Karolina Pawlak.

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
