## [Decision Letter · Decision Letter 0]

PONE-D-24-30213Towards greater integration: Prospects for the development of agri-food trade between the EU and RCEP countriesPLOS ONE

Dear Dr. Bajan,

Thank you for submitting your manuscript to PLOS ONE. After careful consideration, we feel that it has merit but does not fully meet PLOS ONE’s publication criteria as it currently stands. Therefore, we invite you to submit a revised version of the manuscript that addresses the points raised during the review process.

Please submit your revised manuscript by Dec 20 2024 11:59PM. If you will need more time than this to complete your revisions, please reply to this message or contact the journal office at plosone@plos.org . Please include the following items when submitting your revised manuscript:

We look forward to receiving your revised manuscript.

Kind regards,

Anu Sayal, Ph.D.

Academic Editor

PLOS ONE

Journal Requirements:

“This study was founded by the National Science Centre, Poland, under the grant number 2022/47/O/HS4/00548”

3. Please include captions for your Supporting Information files at the end of your manuscript, and update any in-text citations to match accordingly. Please see our Supporting Information guidelines for more information: http://journals.plos.org/plosone/s/supporting-information .

Reviewers' comments:

Reviewer's Responses to Questions

**Comments to the Author**

1. Is the manuscript technically sound, and do the data support the conclusions?

Reviewer #1: Yes

Reviewer #2: Yes

Reviewer #3: Partly

2. Has the statistical analysis been performed appropriately and rigorously? 

Reviewer #1: Yes

Reviewer #2: Yes

Reviewer #3: N/A

3. Have the authors made all data underlying the findings in their manuscript fully available?

Reviewer #1: Yes

Reviewer #2: Yes

Reviewer #3: No

4. Is the manuscript presented in an intelligible fashion and written in standard English?

Reviewer #1: Yes

Reviewer #2: Yes

Reviewer #3: Yes

5. Review Comments to the Author

Reviewer #1: Prof. dr hab. Hanna Klikocka Lublin, 26.10.2024.

University of Life Sciences in Lublin

Manuscript Number: PONE-D-24-30213

Manuscript Title: Towards greater integration: Prospects for the development of agri-food trade between the EU and RCEP countries

The Regional Comprehensive Economic Partnership (RCEP) was concluded in 2020 by ASEAN countries, China, Japan, South Korea, Australia and New Zealand and is the largest free trade area in the world. On the one hand, it is an important trading partner, and on the other hand, it is a competitor to the EU economy, which is striving to maintain its current strong position in international trade.

The study was conducted for the first 24 chapters classified in the Harmonized Commodity Description and Coding System (HS), which includes the agri-food trade.

In order to eliminate one-year fluctuations, the study was conducted for five two-year periods: 2013/2014, 2015/2016, 2017/2018, 2019/2020 and 2021/2022. Were analysed the similarity of agri-food exports between the EU and RCEP.

Firstly, were analysed the similarity of agri-food exports between the EU and RCEP using two indicators, namely the Export Similarity Index (ESI) and the Product Similarity Index (PSI). In the next step, the intensity and nature of intra-industry trade between EU and RCEP countries were calculated. In the next step, the quality changes in trade by dividing intra-industry trade (IIT) into horizontal and vertical trade were analysed.

The study finds stable, moderate competition in agri-food trade between the EU and RCEP, with similarity indices ranging from 0.32 to 0.35. The agri-food trade between the EU and RCEP from 2013 to 2022 intensified. he study provides a strategic framework for future negotiations, emphasizing sectoral approaches to optimize trade outcomes.

The research articles meets the following criteria:

1. The study presents the results of original research.

2. Results reported have not been published elsewhere.

3. Experiments, statistics, and other analyses are performed to a high technical standard and are described in sufficient detail.

4. Conclusions are presented in an appropriate fashion and are supported by the data.

5. The article is presented in an intelligible fashion and is written in standard English.

6. The research meets all applicable standards for the ethics of experimentation and research integrity.

7. The article adheres to appropriate reporting guidelines and community standards for data availability.

Notes:

I suggest that under table 1 you put full nomenclature of the analyzed product groups 1-24).

After these corrections, I recommend publication for publication in a selected journal.

Prof. dr hab. Hanna Klikocka

Reviewer #2: The paper "Towards greater integration: Prospects for the development of agri-food trade between the EU and RCEP countries "presents a comprehensive analysis of agri-food trade potential between the European Union (EU) and the Regional Comprehensive Economic Partnership (RCEP) countries, focusing on trade structure similarity, intra-industry trade, and trade policy implications. The study is scientifically sound, utilizing robust methodologies like the Export Similarity Index (ESI), Product Similarity Index (PSI), Grubel-Lloyd index, and intra-industry trade measures to assess trade dynamics. Its innovative contribution lies in providing a sector-specific, longitudinal examination of trade between these regions, addressing both competition and cooperation potential—a particularly relevant angle given recent geopolitical and economic shifts.

The introduction effectively contextualizes the importance of the EU-RCEP trade relationship and highlights the potential for enhanced agri-food trade integration. It provides a strong rationale, drawing on current trade dynamics and emphasizing the sensitivity of agri-food products in trade negotiations. However, some points, such as specific EU challenges in the global agri-food trade, could benefit from further elaboration to provide a broader perspective.

The literature review is comprehensive, covering relevant studies on regional trade agreements, the dynamics of intra-industry trade, and agri-food trade specifics. It appropriately situates the study within the context of mega-regional trade agreements. The section could improve by incorporating more recent studies to enrich the understanding of evolving trade policy impacts on the EU-RCEP relationship.

The authors detail the methodologies, including the use of ESI, PSI, and intra-industry trade indices, with clarity, ensuring replicability. The data coverage over multiple years strengthens the study’s validity and scope. While methodologically sound, the use of high data disaggregation (6- and 8-digit levels) could be better justified to explain how it enhances the accuracy of trade similarity indices.

Results are presented clearly, with detailed tables and indicators providing insight into EU-RCEP trade patterns. The analysis of the ESI and PSI values is logically structured and offers valuable insights. The section could improve by including graphical representations of trends over time, which would make the longitudinal aspect more accessible and emphasize changes in trade dynamics.

The conclusion effectively summarizes the study's findings, stressing the potential for EU-RCEP cooperation in high-quality agri-food products. It also provides a strong strategic framework that highlights targeted trade policy recommendations. Additional specific recommendations for stakeholders, such as policymakers and industry leaders, could enhance the conclusion's practical relevance.

This study offers valuable insights into EU-RCEP agri-food trade, with potential improvements to strengthen its impact and accessibility for various stakeholders. Despite its value, I believe it would be more robust if it incorporates some recommendations for improvement:

1. Incorporate more recent literature and diverse perspectives, particularly on recent shifts in EU-RCEP relations due to global trade policy changes.

2. Add graphs or charts in the Results section to visually convey trends and make it easier to interpret the longitudinal changes in trade indices.

3. Clarify the advantages of using highly disaggregated data in the methodology section, explaining its impact on the accuracy and relevance of findings.

4. Make specific recommendations for policymakers and industry stakeholders, linking findings directly to practical applications.

5. Discuss potential limitations of the study, such as data availability or specific trade barriers, and suggest avenues for further research.

6. Include a more detailed examination of non-tariff barriers affecting agri-food trade between the EU and RCEP, such as regulatory and quality standards.

Reviewer #3: The theme of this manuscript is very interesting, studying the level of competition in agricultural food trade between the EU and RCEP and the best areas for strengthening trade cooperation. However, there are still many issues with the manuscript.

Abstract

* The research contributions of the paper should be articulated more clearly. The abstract is not representative of the content and contributions of the paper. The abstract does not seem to properly convey the rigor of research.

* Aside from the aim stated in the title, the research gap and the goals of the research are not specified which leads to the reader missing the significance of the research.

Introduction

* The introduction section is detailed, but needs a significant amount of reorganization. It could be strengthened by adding more recent references.

* Please add as sentence or two to clearly recap how your study differs from what has already been done in literature to ascertain the contributions more strongly

* More explanation is needed for where there is a research gap and what the goals of the research are. The research gap and the goals of the research are not explained in detail which leads to the reader missing the significance of the research.

* The research idea should be linked to multiple problems the research is trying to address so that the findings have relevance.

Literature Review

* Sources are out of date. More recent studies (2021-2024) should be included. Also, it should lead up to the research questions in a logical manner.

* Please discuss learning theories and tie them to both, the research gap addressed by the paper as well as to the factors in the research model.

* The literature review is insufficient in its addressal of the research gap and research model.

Methodology

* The methodology section needs more details and a drastic revision.

* I suggest using regression analysis.

* The subsections of participants, instruments, data collection procedures, and data analysis should be separately given

* The items in the instrument used, demographic information, reliability and validity information, any statistical or data analysis should be presented in detail.

* The reason for using specific analysis is not clearly mentioned. Justification for using a specific methodology or instrument will make it more understandable. Adding more details in this section can give more clarity to the readers

* The methodology used should be justified in the article in the light of the research questions (i.e. why is the chosen methodology the best approach to answer the research questions).

Discussion

* Improve the discussion section to better ascertain what is unique / novel about your findings

* Explain in detail how the article contributes to new knowledge in the domain.

* The manuscript lacks policy recommendations.

Conclusion

* Update the conclusion to include the newly formulated theoretical contributions

* Mention the limitations of the study and prospects for future research.

* Summarize the key results in a compact form and re-emphasize their significance.

* Summarize how the article contributes to new knowledge in the domain.

6. PLOS authors have the option to publish the peer review history of their article (what does this mean? ). If published, this will include your full peer review and any attached files.

**Do you want your identity to be public for this peer review?** For information about this choice, including consent withdrawal, please see our Privacy Policy .

Reviewer #1: No

Reviewer #2: **Yes: ** Pedro Migueel Magalhães Nunes Chamusca

Reviewer #3: No

---

## [Author Response · Author response to Decision Letter 1]

9 Dec 2024

We would like to thank the Reviewers for the feedback and all the comments that helped us improve the paper. Below we explain the changes made in line with the reviewer’s recommendations. We believe that the revisions improved the quality and scientific soundness of the paper. All additions can be seen in the file with tracked changes.

Reviewer #1

Recommendation:

1. I suggest that under table 1 you put full nomenclature of the analyzed product groups 1-24).

According to your suggestion, a full nomenclature of the analyzed product groups was inserted under Table 1.

Reviewer #2

Recommendations:

1. Incorporate more recent literature and diverse perspectives, particularly on recent shifts in EU-RCEP relations due to global trade policy changes.

We incorporated a more specific description of EU challenges due to global trade policy changes, particularly focusing on Asia-Pacific relations based on recent literature. An appropriate part covering the EU strategy to pursue FTAs with several Indo-Pacific nations as a counterbalance to RCEP's influence was added to the introduction. Moreover, risks for the EU posed by the US-China trade war were articulated, and potential opportunities for the EU to leverage its partnerships in Asia were mentioned.

2. Add graphs or charts in the Results section to visually convey trends and make it easier to interpret the longitudinal changes in trade indices.

The trend lines were added to Figure 1 (results of ESI and PSI). Moreover, an additional figure was added to better visualize changes in intra-industry trade.

3. Clarify the advantages of using highly disaggregated data in the methodology section, explaining its impact on the accuracy and relevance of findings.

According to your suggestion, we have developed the validity of using disaggregated data. Previously, information about this was included in two places in the methodology:

“We aimed for the highest possible level of disaggregation as it significantly impacts the values of the indicators we used [31]. As the level of data disaggregation increases, their values tend to decrease, giving more accurate estimation [32].”

“Using highly disaggregated data can help conduct an in-depth analysis and obtain more information on the import and export patterns of the countries, which allows a better understanding of trade dynamics [35].”

We also added an additional explanation:

“Disaggregated data enables the study to identify trade patterns specific to individual product categories, leading to more accurate measurements of trade similarity and intra-industry trade levels. By tracking exports and imports at the 6- and 8-digit HS code level, the analysis can capture variations across a broad range of agri-food products, enhancing the specificity and relevance of the results [36].”

4. Make specific recommendations for policymakers and industry stakeholders, linking findings directly to practical applications.

We created a more detailed recommendation, suggesting specific actions and their effects, with the goal of increasing trade between the EU and RCEP.

5. Discuss potential limitations of the study, such as data availability or specific trade barriers, and suggest avenues for further research.

In the description, we added limitations to our study, where we focused on non-tariff barriers to trade between EU and RCEP countries. In addition, we described in more detail the possibilities for future research.

6. Include a more detailed examination of non-tariff barriers affecting agri-food trade between the EU and RCEP, such as regulatory and quality standards.

In line with your recommendation, in the results and discussion section, we paid more attention to non-tariff barriers affecting trade in agri-food products between the EU and RCEP.

Reviewer #3

Recommendations:

Abstract

1. The research contributions of the paper should be articulated more clearly. The abstract is not representative of the content and contributions of the paper. The abstract does not seem to properly convey the rigor of research.

We have revised the abstract to clearly articulate the paper’s contributions. The updated abstract now emphasizes the study’s primary focus on identifying opportunities for sectoral cooperation and competitive dynamics within EU-RCEP agri-food trade, addressing an acknowledged gap in existing research. By specifically highlighting findings such as stable but moderate competition and the potential for EU exports of high-quality agri-food products, we aim to showcase both the significance and the depth of our analysis. The revised abstract provides a clearer representation of the paper's objectives, contributions, and results.

2. Aside from the aim stated in the title, the research gap and the goals of the research are not specified which leads to the reader missing the significance of the research.

In the revised abstract, we explicitly outline the research gap - the lack of detailed analysis of EU-RCEP agri-food trade relations. The revised version now states that the goal of the study is to assess sectoral cooperation opportunities and competitive dynamics, establishing a foundation for potential future trade negotiations. These additions underscore the study's purpose and relevance, allowing readers to understand better the strategic significance of our research in shaping EU-RCEP trade policy.

Introduction

1. The introduction section is detailed, but needs a significant amount of reorganization. It could be strengthened by adding more recent references.

More recent references have been added, as suggested. After these changes, the introduction uses 21 literature sources, of which more than half were published in the last 5 years and 8 of them in the last 3 years. Moreover, the introduction was reorganized by adding a paragraph on recent policy shifts affecting EU-RCEP relations. Furthermore, more emphasis was placed on the novelty of the study.

2. Please add as sentence or two to clearly recap how your study differs from what has already been done in literature to ascertain the contributions more strongly

According to this suggestion, we added the following sentences: “The research so far is concerned solely with macro-level relations between two groupings or relations between the EU and single countries in the Asia-Pacific region. Sectoral-level studies, to date, are of a descriptive nature or covering only a few groups of products”.

3. More explanation is needed for where there is a research gap and what the goals of the research are. The research gap and the goals of the research are not explained in detail which leads to the reader missing the significance of the research.

According to this suggestion, we added the following sentences: “the novelty of our research lies in a detailed analysis of agri-food trade between the EU and RCEP, covering 24 chapters of agri-food products.”, and “we aim to analyse trade potential for each product category between the EU and RCEP based on empirical data. Therefore extending the current state of knowledge through evidence-based insides”.

4. The research idea should be linked to multiple problems the research is trying to address so that the findings have relevance.

This issue was solved by adding a paragraph-length description of recent shifts in trade policies and their link to the EU-RCEP situation. We mentioned EU strategy to pursue FTAs with several Indo-Pacific nations as a counterbalance to RCEP's influence. Moreover, risks for the EU posed by the US-China trade war were articulated, and potential opportunities for the EU to leverage its partnerships in Asia were mentioned. “For the EU, these developments pose a dual challenge: leveraging its bilateral agreements with RCEP members to secure market access while navigating the effects of large-scale initiatives of the US and China”. The issues mentioned correspond directly with our research idea to “map the baseline situation for potential negotiations of FTA between the EU and RCEP regarding agri-food products”.

Literature Review

1. Sources are out of date. More recent studies (2021-2024) should be included. Also, it should lead up to the research questions in a logical manner.

The literature review has been extended to include more recent literature on the subject. The novelty of the study and theory-based justification for the analyses performed were also emphasized. As a result, the research gap is better addressed, and the study’s contributions are clarified in more detail.

2. Please discuss learning theories and tie them to both, the research gap addressed by the paper as well as to the factors in the research model.

Thank you for this comment. Now, the literature review section discusses theories that serve as a background for the analyses performed. The theory of comparative advantage, the theory of overlapping demand, the theory of intra-industry trade, and the customs union theory were considered to address the research gap and show the study’s importance.

3. The literature review is insufficient in its addressal of the research gap and research model.

In line with the reviewer’s recommendation, the literature review has been extended to clarify the research gap and the study’s contributions in more detail. We took advantage of more recent studies and employed a few theories, which constitute the foundations for the analyses performed.

Methodology

1. The methodology section needs more details and a drastic revision.

We introduced more details in the methodology. We have added information about the rationale for using disaggregated data, and we have also focused on how the chosen methods fit our research questions. In addition, we divided the methodology into sections.

2. I suggest using regression analysis.

We agree that such an addition could be beneficial. However, we treat it as an idea for different studies, as it is not crucial for fulfilling our goal. We included the idea of using regression analysis as a future direction of research. A regression analysis or gravity model can definitely help assess the strength of possible effects (trade shifting and creation) on trade between the EU and RCEP.

3. The subsections of participants, instruments, data collection procedures, and data analysis should be separately given.

According to this remark, we restructured this section, splitting it into “Data Collection” and “Calculation Procedure”, to boost readability.

4. The items in the instrument used, demographic information, reliability and validity information, any statistical or data analysis should be presented in detail.

Additional information has been added to the Materials and Methods section. We focused on further clarifying the advantages of using highly disaggregated data and explaining its impact on the accuracy and relevance of results. Moreover, we explicitly gave the number of countries considered in the analysis. Furthermore, the number of commodities analyzed under the 8-digit HS classification was provided.

5. The reason for using specific analysis is not clearly mentioned. Justification for using a specific methodology or instrument will make it more understandable. Adding more details in this section can give more clarity to the readers.

In line with your recommendation, we have added more details about the methods, as well as a justification for choosing a particular method of analysis.

6. The methodology used should be justified in the article in the light of the research questions (i.e. why is the chosen methodology the best approach to answer the research questions).

In the data and methods section, we have also added an explanation of how the chosen methods will answer the research questions.

Discussion

1. Improve the discussion section to better ascertain what is unique / novel about your findings.

In accordance with your recommendation, both in the introduction and in the discussion, we drew attention to the novelty of our article.

2. Explain in detail how the article contributes to new knowledge in the domain.

We have added an explanation of how our study creates new knowledge in the domain.

3. The manuscript lacks policy recommendations.

We created a more detailed recommendation, suggesting specific actions and their effects, with the goal of increasing trade between the EU and RCEP.

Conclusion

1. Update the conclusion to include the newly formulated theoretical contributions.

Following your recommendation, we have formulated what contribution our article makes to theory. The following sentence was added: “This study contributes to the theoretical discourse on regional trade integration by illustrating how structural trade differences in agri-food products affect intra-industry trade, by empirically linking quality differentiation (e.g., VIIThigh and VIITlow) to trade potential.”

2. Mention the limitations of the study and prospects for future research.

In the description, we added limitations to our study, where we focused on non-tariff barriers to trade between EU and RCEP countries. In addition, we described in more detail the possibilities for future research. The following sentences were added “. Conversely, a limitation of our study is the failure to measure a specific effect of non-tariff barriers, such as sanitary and phytosanitary regulations and environmental regulations. Restrictive EU standards for food quality, health protection, and CO₂ emissions do not always align with those of RCEP countries, resulting in the need for additional certification and product adjustments, which can hinder trade.” “Future studies should also consider non-tariff barriers and their impact on trade. For this purpose, it would be possible to use regression analysis or gravity model.”

3. Summarize the key results in a compact form and re-emphasize their significance.

We have added key findings to the summary: “Therefore, key findings include, firstly, moderate competition in agri-food trade between the EU and RCEP, and secondly, the dominance of high-quality vertical trade flows (VIIThigh) suggests the EU's competitive advantage in high-value agri-food products. We also observed weak but strengthening intra-industry trade in select product groups, which highlights untapped potential for trade cooperation. These insights underscore the need for harmonized strategic export policies to develop trade relations between EU and RCEP countries.”

4. Summarize how the article contributes to new knowledge in the domain.

In the conclusions, we added what new knowledge the article brings to the domain. We focused mainly on structural differences that affect intra-industry trade and differences in product quality that affect the potential for trade development.

---

## [Decision Letter · Decision Letter 1]

PONE-D-24-30213R1Towards greater integration: Prospects for the development of agri-food trade between the EU and RCEP countriesPLOS ONE

Dear Dr. Bajan,

Thank you for submitting your manuscript to PLOS ONE. After careful consideration, we feel that it has merit but does not fully meet PLOS ONE’s publication criteria as it currently stands. Therefore, we invite you to submit a revised version of the manuscript that addresses the points raised during the review process.

The manuscript has been evaluated by four reviewers, and their comments are provided below. One reviewer raised several concerns that require attention. Specifically, they requested additional details in the methods section to enhance the reproducibility of this work. They also had concerns related to the lack of literature on Geographical Indications (GI) and suggested minor corrections in various parts of the manuscript.

Could you please carefully revise the manuscript to address all comments raised?

We look forward to receiving your revised manuscript.

Kind regards,

Zahra Al-Khateeb, Ph.D

Staff Editor

PLOS ONE

Reviewers' comments:

Reviewer's Responses to Questions

**Comments to the Author**

1. If the authors have adequately addressed your comments raised in a previous round of review and you feel that this manuscript is now acceptable for publication, you may indicate that here to bypass the “Comments to the Author” section, enter your conflict of interest statement in the “Confidential to Editor” section, and submit your "Accept" recommendation.

Reviewer #1: All comments have been addressed

Reviewer #2: All comments have been addressed

Reviewer #4: (No Response)

Reviewer #5: All comments have been addressed

2. Is the manuscript technically sound, and do the data support the conclusions?

Reviewer #1: Yes

Reviewer #2: Yes

Reviewer #4: Partly

Reviewer #5: Yes

3. Has the statistical analysis been performed appropriately and rigorously? 

Reviewer #1: Yes

Reviewer #2: Yes

Reviewer #4: N/A

Reviewer #5: Yes

4. Have the authors made all data underlying the findings in their manuscript fully available?

Reviewer #1: Yes

Reviewer #2: Yes

Reviewer #4: Yes

Reviewer #5: Yes

5. Is the manuscript presented in an intelligible fashion and written in standard English?

Reviewer #1: Yes

Reviewer #2: Yes

Reviewer #4: Yes

Reviewer #5: Yes

6. Review Comments to the Author

Reviewer #1: The work presented for evaluation is interesting. The authors made the corrections correctly. The methodological part is correct. The selection of literature and its list are correct. The literature review, discussion and summary are correct. Therefore, in this version I recommend the work for printing.

Reviewer #2: All the comments have been addressed by the authors. The text can be published, considering the research is scientifically interesting and relevant.

Reviewer #4: This paper conducts a descriptive analysis of trade similarity and horizontal/vertical intra-industry trade between the EU and the RCEP countries. It delivers on what it claims, but the analysis remains very descriptive. In addition, essential procedural information/interpretation steps are missing.

I was asked to review a revision but have not reviewed the original version, so apologies if some of my comments are new compared to the first round of reviews.

Major comments

Major comment 1 – issues with Table 1 and the main analysis

1a. The methodology description is insufficient. Who is the exporter and who is the importer? Since it matters for saying whether the trade is high vertical or low vertical. From the discussion RCEP appears to be the importer, but that should be made explicit in the formulas (e.g. equation 6) and in the table heading.

1b. How did you collapse this from CN8 to HS2 headings? Probably you took the average, but essential things like this should be explained for reproducibility.

1c. In the conclusion you write “high-quality vertical trade was observed in most product groups” - This statement may be true but requires more explicit analysis. Not even a percentage of HS2/HS6 headings for which this is the case is provided in the text.

Major comment 2

Given the repeated references to EU exports of high-quality goods, one would have expected literature on Geographical Indications (GI) to be cited. These protected regional foods have been shown to have higher unit values, and their protection plays a key role in trade negotiations. By asking trading partners to protect them, the EU hopes/aims to export even more of its high-quality products, as imitations are taken off the market. The abstract has the sentence “While intra-industry trade remains limited due to geographic distance, there is notable potential for expanding high-quality EU agri-food exports” – the legal protection of GIs in RCEP countries (to the extend they are not already subject to a bilateral FTA like e.g. Japan is) would be a channel for that.

Minor comments

Line 97-98: Did you check relevant policy documents, e.g. the EU Impact Assessment of the EU-Indonesia FTA available at https://policy.trade.ec.europa.eu/analysis-and-assessment/sustainability-impact-assessments_en It definitely has relevant findings, e.g. on p. 9 "For agri-foods, the model predicts that the agreement could lead to increased production of dairy and alcoholic beverages in the EU"

Line 216 “trade in agri-food products as being one of the most sensitive when negotiating trade liberalization” – indeed, all the more reason to cite literature on GIs. Their protection by trading partners is often seen as counterweight for increased market access into EU market.

Line 248 “8-digit HS”; technically this is CN8, since the HS is worldwide, but beyond HS6 countries use their own systems (and the EU uses CN8)

Line 262 “c denotes shares of exported goods in the total agri-food export”- you mean x denotes this? And c is the index of goods from 1 to n?

Line 275 you need to add “of Intra-industry trade (IIT)” to the Gruber-Lloyd index

Line 316 eqn7 - give this a different name than (3), since the formula is different

Reviewer #5: (No Response)

7. PLOS authors have the option to publish the peer review history of their article (what does this mean? ). If published, this will include your full peer review and any attached files.

**Do you want your identity to be public for this peer review?** For information about this choice, including consent withdrawal, please see our Privacy Policy .

Reviewer #1: No

Reviewer #2: No

Reviewer #4: No

Reviewer #5: No

---

## [Author Response · Author response to Decision Letter 2]

16 Apr 2025

We would like to thank the Reviewer for the feedback and all the comments that helped us improve the paper. Below we explain the changes made in line with the reviewer’s recommendations. We believe that the revisions improved the quality and scientific soundness of the paper. All additions can be seen in the file with tracked changes.

Recommendations:

Major comment 1 – issues with Table 1 and the main analysis

1a. The methodology description is insufficient. Who is the exporter and who is the importer? Since it matters for saying whether the trade is high vertical or low vertical. From the discussion RCEP appears to be the importer, but that should be made explicit in the formulas (e.g. equation 6) and in the table heading.

We agree that it was not explicitly stated when describing the methodology that we are analysing intra-industry trade from an EU perspective, so when describing the indicators for horizontal and vertical trade, we have indicated that we are referring to the EU.

We have also made a change to the name of the table.

1b. How did you collapse this from CN8 to HS2 headings? Probably you took the average, but essential things like this should be explained for reproducibility.

A detailed explanation has been added to the Materials and Methods section: “In the analysis of intra-industry trade, the CN classification at the 8-digit level was used, which is a more detailed extension of the HS classification. This allows each CN code to be assigned to the appropriate product group at the 2-digit HS level. Ratios of unit values of exports (UVX) to imports (UVM) are calculated at the level of 8-digit CN codes. Based on the results for individual CN codes, the value of products traded within horizontal and vertical trade can then be determined for each product group classified at the 2-digit HS level. This approach makes it possible to obtain aggregated indicators for each general product group at the 2-digit HS level. In other words, the analysis reveals the value of products within a given group (2-digit HS) that constitutes a specific type of intra-industry trade, based on classifications made at the detailed CN level”.

1c. In the conclusion you write “high-quality vertical trade was observed in most product groups” - This statement may be true but requires more explicit analysis. Not even a percentage of HS2/HS6 headings for which this is the case is provided in the text.

In both the description of the results and the summary, a clarification was added as to the share of high-quality products in intra-industry trade between EU and RCEP countries.

Major comment 2

Given the repeated references to EU exports of high-quality goods, one would have expected literature on Geographical Indications (GI) to be cited. These protected regional foods have been shown to have higher unit values, and their protection plays a key role in trade negotiations. By asking trading partners to protect them, the EU hopes/aims to export even more of its high-quality products, as imitations are taken off the market. The abstract has the sentence “While intra-industry trade remains limited due to geographic distance, there is notable potential for expanding high-quality EU agri-food exports” – the legal protection of GIs in RCEP countries (to the extend they are not already subject to a bilateral FTA like e.g. Japan is) would be a channel for that.

Thank you for this particularly useful remark! A paragraph length explanation with literature on Geographical Indications has been added to the discussion, linking our findings with other studies and already existing GI provisions between the EU and RCEP countries.

Minor comments

Line 97-98: Did you check relevant policy documents, e.g. the EU Impact Assessment of the EU-Indonesia FTA available at https://policy.trade.ec.europa.eu/analysis-and-assessment/sustainability-impact-assessments_en It definitely has relevant findings, e.g. on p. 9 "For agri-foods, the model predicts that the agreement could lead to increased production of dairy and alcoholic beverages in the EU"

Thank you for pointing out this omission. The following explanation has been added to the introduction: “Some relevant policy assessments–Sustainability Impact Assessments–of ongoing negotiations exist [22]; however, these studies are limited to bilateral contexts. We aim to map a comprehensive view of the potential of agri-food trade between the EU and RCEP groupings”. Moreover, the discussion has been enhanced by some provisions of the Sustainability Impact Assessment of the EU–Indonesia Free Trade Agreement.

Line 216 “trade in agri-food products as being one of the most sensitive when negotiating trade liberalization” – indeed, all the more reason to cite literature on GIs. Their protection by trading partners is often seen as counterweight for increased market access into EU market.

This comment has been covered by changes based on previous remarks.

Line 248 “8-digit HS”; technically this is CN8, since the HS is worldwide, but beyond HS6 countries use their own systems (and the EU uses CN8)

Changes have been made and it has been explained what the CN8 classification is in the Materials and Methods section.

Line 262 “c denotes shares of exported goods in the total agri-food export”- you mean x denotes this? And c is the index of goods from 1 to n?

This misunderstanding has been clarify by stating that x indicates shares of exported goods in the total agri-food export, c denotes exported commodity.

Line 275 you need to add “of Intra-industry trade (IIT)” to the Gruber-Lloyd index

A change has been made, as you have suggested.

Line 316 eqn7 - give this a different name than (3), since the formula is different

Following your recommendation, the name of the index has been changed. It is now the Quality-Differentiated IIT index to emphasize that it analyses intra-industry trade by vertical and horizontal trade, taking into account qualitative differences.

---

## [Decision Letter · Decision Letter 2]

PONE-D-24-30213R2Towards greater integration: Prospects for the development of agri-food trade between the EU and RCEP countriesPLOS ONE

Dear Dr. Bajan,

Thank you for submitting your manuscript to PLOS ONE. After careful consideration, we feel that it has merit but does not fully meet PLOS ONE’s publication criteria as it currently stands. Therefore, we invite you to submit a revised version of the manuscript that addresses the points raised during the review process.

We look forward to receiving your revised manuscript.

Kind regards,

Jian Xu

Academic Editor

PLOS ONE

Additional Editor Comments:

Please revise the paper according to the comments of the reviewers.

Reviewers' comments:

Reviewer's Responses to Questions

**Comments to the Author**

1. If the authors have adequately addressed your comments raised in a previous round of review and you feel that this manuscript is now acceptable for publication, you may indicate that here to bypass the “Comments to the Author” section, enter your conflict of interest statement in the “Confidential to Editor” section, and submit your "Accept" recommendation.

Reviewer #1: All comments have been addressed

Reviewer #5: All comments have been addressed

2. Is the manuscript technically sound, and do the data support the conclusions?

Reviewer #1: Yes

Reviewer #5: Yes

3. Has the statistical analysis been performed appropriately and rigorously? 

Reviewer #1: Yes

Reviewer #5: Yes

4. Have the authors made all data underlying the findings in their manuscript fully available?

Reviewer #1: Yes

Reviewer #5: Yes

5. Is the manuscript presented in an intelligible fashion and written in standard English?

Reviewer #1: Yes

Reviewer #5: Yes

6. Review Comments to the Author

Reviewer #1: I have no comments on the manuscript. The correction was carried out correctly. I suggest to print it.

Reviewer #5: (No Response)

7. PLOS authors have the option to publish the peer review history of their article (what does this mean? ). If published, this will include your full peer review and any attached files.

**Do you want your identity to be public for this peer review?** For information about this choice, including consent withdrawal, please see our Privacy Policy .

Reviewer #1: No

Reviewer #5: No

---

## [Author Response · Author response to Decision Letter 3]

30 Jun 2025

I was informed by e-mail that , a Revise decision, was issued in error and as such, I should resubmit the paper without revision.

---

## [Editor Report · Decision Letter 3]

Towards greater integration: Prospects for the development of agri-food trade between the EU and RCEP countries

PONE-D-24-30213R3

Dear,

We’re pleased to inform you that your manuscript has been judged scientifically suitable for publication and will be formally accepted for publication once it meets all outstanding technical requirements.

Kind regards,

Jian Xu

Academic Editor

PLOS ONE

Additional Editor Comments (optional):

I think the paper can be accepted in its current form.
---

## [Editor Report · Acceptance letter]

PONE-D-24-30213R3

PLOS ONE

Dear Dr. Bajan,

I'm pleased to inform you that your manuscript has been deemed suitable for publication in PLOS ONE. Congratulations! Your manuscript is now being handed over to our production team.

Kind regards,

on behalf of

Dr. Jian Xu

Academic Editor

PLOS ONE